# Cell-Penetrating Peptides: Applications in Tumor Diagnosis and Therapeutics

**DOI:** 10.3390/pharmaceutics13060890

**Published:** 2021-06-15

**Authors:** Jeffrey Stiltner, Kayla McCandless, Maliha Zahid

**Affiliations:** Rangos Research Center, Department of Developmental Biology, University of Pittsburgh School of Medicine, 530 45th Street, Pittsburgh, PA 15201, USA; JES362@pitt.edu (J.S.); KAM514@pitt.edu (K.M.)

**Keywords:** cell-penetrating peptides, protein transduction domains, tumor imaging, targeted therapies

## Abstract

Since their identification over twenty-five years ago, the plethora of cell-penetrating peptides (CPP) and their applications has skyrocketed. These 5 to 30 amino acid in length peptides have the unique property of breaching the cell membrane barrier while carrying cargoes larger than themselves into cells in an intact, functional form. CPPs can be conjugated to fluorophores, activatable probes, radioisotopes or contrast agents for imaging tissues, such as tumors. There is no singular mechanism for translocation of CPPs into a cell, and therefore, many CPPs are taken up by a multitude of cell types, creating the challenge of tumor-specific translocation and hindering clinical effectiveness. Varying strategies have been developed to combat this issue and enhance their diagnostic potential by derivatizing CPPs for better targeting by constructing specific cell-activated forms. These methods are currently being used to image integrin-expressing tumors, breast cancer cells, human histiocytic lymphoma and protease-secreting fibrosarcoma cells, to name a few. Additionally, identifying safe, effective therapeutics for malignant tumors has long been an active area of research. CPPs can circumvent many of the complications found in treating cancer with conventional therapeutics by targeted delivery of drugs into tumors, thereby decreasing off-target side effects, a feat not achievable by currently employed conventional chemotherapeutics. Myriad types of chemotherapeutics such as tyrosine kinase inhibitors, antitumor antibodies and nanoparticles can be functionally attached to these peptides, leading to the possibility of delivering established and novel cancer therapeutics directly to tumor tissue. While much research is needed to overcome potential issues with these peptides, they offer a significant advancement over current mechanisms to treat cancer. In this review, we present a brief overview of the research, leading to identification of CPPs with a comprehensive state-of-the-art review on the role of these novel peptides in both cancer diagnostics as well as therapeutics.

## 1. Introduction

As so often happens in science, the discovery of cell-penetrating peptides (CPP) was a serendipitous one. Two independent groups of researchers working on the human immunodeficiency virus (HIV) viral coat Trans-activator of Transcription (Tat) protein observed the protein’s ability to cross cell membrane barriers without any transfection reagents [1,2]. Similarly, the homeobox Antennapedia (Antp) transcription factor of *Drosophila melanogaster* was demonstrated to enter nerve cells in a receptor-independent manner, where it could then regulate neural morphogenesis [3]. Further mapping studies of the domains within Tat and Antp responsible for the observed transduction led to the identification of the first two CPPS: the 11 amino acid cationic, arginine- and lysine-rich domain of Tat protein (YGRKKRRQRRR) [4] and the 16 amino acid sequence from the third helix of the Antennapedia domain (RQIKIWFQNRRMKWKK) termed Antp, also known as penetratin [5]. The next big development in the field of CPPs came with the demonstration of Tat peptide’s ability to cross cell membrane barriers while carrying cargo many times its size in a functional form [6]. Since this initial description, the plethora of CPPs has expanded exponentially. Although the first two CPPs identified were non-cell specific, researchers have utilized phage-display methodologies to identify multiple tissue-specific peptides. Phage display was a technique developed by Smith in 1985 [7], and for which he subsequently received the Nobel prize for chemistry in 2018 [8]. The technique of phage display was initially utilized to identify NRG and RGD motifs targeting tumor cells, and the utility of these peptides in delivering chemotherapeutic agents specifically to tumor vasculature was demonstrated [9]. Phage display has also been used to identify peptides targeting vascular endothelium [10], synovial tissue [11], dendritic cells [12], pancreatic islet cells [13] and cardiac myocytes [14]. Additionally, this list continues to grow every year. Hence, no one review article can do full justice to the entire breadth of CPPs, tissue and non-tissue selective, their myriad cargoes, and the number of disease conditions being tackled using them. Therefore, out of necessity, this review will be limited to tumor-homing CPPs, and utility of these in tumor imaging and tumor-specific therapeutics. Interested readers are referred to several recent comprehensive reviews on other uses of CPPs [15,16].

## 2. Cell-Penetrating Peptides as Tumor Imaging Agents

CPPs are a promising tool for tumor imaging due to their high binding affinity, small size, specific uptake, high stability, rapid clearance from non-specific targets, and retention in specific targets [17,18,19,20,21]. They can be conjugated to radioisotopes, fluorophore-labeled or activatable probes, nanoparticles (NPs), polymers, quantum dots, metal chelates, and other contrast agents in order to image tumors [22,23,24,25,26,27]. CPPs are able to carry, transport, and deliver these imaging agents, providing the imaging cargo with intracellular access and functionality. Since every CPP is different and has varying chemical properties due to differences in their amino acid sequence, each faces its own challenges. An additional layer of complexity comes from the cargo it carries as that too can affect the chemical properties. Therefore, it is always important to assess the short comings of each CPP individually and when loaded with its cargo [28,29]. Some challenges to using CPPs for tumor imaging include serum stability, immunogenicity, cytotoxicity, and endosomal entrapment [30,31]. There is also no singular mechanism for translocation of CPPs into a cell, and therefore many CPPs are taken up by a multitude of cell types, creating the challenge of tumor-specific translocation and hindering clinical effectiveness [32]. Various strategies are currently being developed to combat these issues and enhance tumor diagnostic imaging. Some examples include selecting CPPs for their targeting abilities or labeling CPPs with specific cell-activated constructs [33,34]. One of the strategies is to select CPPs to image cancer tissues by taking advantage of overexpression of integrins by tumors, as seen in breast cancer, human histiocytic lymphoma U937, HT-1080 human fibrosarcoma cells, and SCC-7 tumors, to name a few [35,36,37].

Non-tumor-targeting CPPs have nevertheless been used for imaging tumors. Although Tat cannot specifically target tumors, its stability in vivo and rapid translocation across cell membranes show promising abilities as peptide chelates and fluorophore conjugates to serve as imaging agents (Figure 1). One study labeled the Tat peptide with technetium-^99m^ (^99m^Tc), one of the most common radioisotopes for medical imaging. The peptide was synthesized using two functional domains, the first being the non-cell-specific, membrane permeant portion of the Tat protein (Tat peptide), and the second domain using a peptide-based chelator for ^99m^Tc (ε-KGC). The [^99m^Tc]-Tat-peptide combination was imaged in mice using a gamma scintillation camera and, as expected, showed whole-body distribution [38]. Another study also used non-specific CPPs labeled with fluorophores instead of isotopes. The CPPs studied consisted of Tat, penetratin (Pen), or octa-arginine L-enantiomer (R8). Each CPP was synthesized with a cysteine or glycyl cysteine amide and labeled with Alexa660 at the C-terminus, injected into HeLa xenografted into nude mice and imaged using an IVIS Spectrum System. The accumulation of R8 in tumors was significantly higher than all other CPPs. Since the number of arginine residues affects internalization efficiency, and D-enantiomers decrease degradation by proteases, the L- and D-forms of the oligoarginines (2, 8, 12, and 16 mers) were repeated. The results showed the D-isoform of R8 as having the highest accumulation in tumors. This study not only highlights CPPs as promising agents for tumor imaging, but also demonstrates how changes in peptide amino acid sequence and configuration can affect their physicochemical properties, leading to a CPP better suited to the task of imaging [39].

A separate strategy to enhance tumor targeting is by dual targeting. In this method, the CPP is conjugated with another agent to increase its targeting ability (Figure 1). Huang et al. synthesized linear RGERPPR (RGE) and cyclic-peptide CRGDRGPDC (cRGD) due to their demonstrated higher affinity for multiple tumor cell lines. The linear RGE and cRGD were conjugated to a lipid carrier in order to enhance cell uptake and tumor targeting. The lipid carrier was embedded with the fluorescent dye DiR for tumor imaging. MDA-MB-231 breast tumor xenograft mouse models were injected with the conjugate and imaged using a near-infrared fluorescence imaging system. The linear RGE conjugate showed the highest uptake and retention in tumors, making RGE and the CPP-NP combination promising tools for tumor imaging [40]. Another study used the CPP and NP dual targeting combination to target tumor cells, but with the addition of another imaging agent for dual-modality imaging (Figure 1) (photoacoustic (PA) and MRI imaging). In this study, the CPP, F3 peptide, and NP, poly(lactic-co-glycolic acid) (PLGA), were used for their cell targeting and penetrating abilities, respectively. The sonosensitizer, methylene blue, was embedded in Gd-DTPA-BMA-linked-PLGA to combine the new imaging modality, PA, with the current clinical imaging modality MRI. MDA-MB-231 tumor-bearing mice were injected with the synthesized F3-PLGA@MB/Gd NP and evaluated using dual-imaging. The results showed that F3-PLGA@MB/Gd NP had the highest concentration in tumors compared to the non-targeted groups in both PA and MRI, most notably at 6 h post-injection. Again, this study highlights a promising CPP (F3) and CPP-NP combination for tumor imaging, demonstrating that there are multiple variations of CPPs and combinations that can be used to advance tumor imaging [41].

Another strategy to enhance tumor imaging is via use of activatable CPPs (ACPPs). In this method, the CPP contains a region which can penetrate cells and carry cargo (polycation), a region which can target metalloproteases-2 (MMP-2) and MMP-9 (protease-cleavable linker), and a region which quenches the function of the cell-penetrating region (polyanion). When activated by MMP-2 and MMP-9, the neutralizing region is cleaved and the CPP can enter the cell (Figure 1). Since MMP-2 and MMP-9 are over expressed in many cancer lines, the hypothesis was that ACPPs will preferentially target tumors over non-tumor tissues. A recent study used an ACPP labeled with Cy5 to target and image colorectal cancer. In vivo and ex vivo fluorescent imaging was performed using an IVIS imaging system in HCT-116 xenograft nude mice. The results showed that ACPP-Cy5 accumulated in tumors, and liver metastases, making it a promising imaging agent for detecting tumors as well as metastases [42]. Specific cargo can also be attached to ACPPs to enhance tumor targeting and imaging abilities. Macromolecules can increase circulation time and tumor uptake, decrease background noise through less glomerular and synovial filtration, amplify the amount of imaging agent on a single peptide, decrease toxicity, and allow for multimodality imaging. Olson et al. conjugated polyamidoamine dendrimer to the polycationic domain of an ACPP. Using Cy5 and Gd chelates allowed for dual labeling of the ACPP dendrimer combination for fluorescent and MRI imaging in tumor-bearing mice. The results showed better uptake and tumor specificity than previously reported results, once again illustrating the myriad ways in which CPPs can be altered and combined to improve tumor imaging [43].

## 3. Cell-Penetrating Peptides as Vectors for Targeted Drug Delivery to Tumors

Cancer is a leading cause of mortality globally, second only to cardiovascular diseases [44]. In 2017, cancer claimed the lives of nearly 10 million people worldwide [44]. As the average life expectancy, standard of living and access to healthcare increase, people are living longer lives, with consequently a shift in mortality rates from infectious diseases [45] to a rise in cancer-related mortality that is predicted to increase globally over the coming decades. Despite advancements in oncological research and medicine, conventional chemotherapeutics still have many limitations and deficiencies. One example is doxorubicin, a common chemotherapeutic agent, which suffers from poor tumor penetrance [46]. This poor penetration translates into deeper areas of the tumor not receiving adequate drug concentrations allowing cancer cells to remain viable and continue to mutate and proliferate [46]. A second issue is the high interstitial pressure present in tumors which blocks efficient delivery of drugs through transcapillary transport [47,48]. This elevated interstitial pressure causes a radially outward pressure away from the tumor, making access by chemotherapeutics through simple diffusion challenging [48]. Another issue is development of tumor resistance to chemotherapy over time [49], the mechanism of which is not well understood, but is thought to involve cancer stem cells playing a role, leading to tumor relapses [50]. Another issue with modern cancer drugs are the large doses needed due to lack of targeting specificity [51,52], which contributes substantially to toxicity and side effects, making chemotherapy poorly tolerated by patients [53].

Cell-penetrating peptides have shown great preclinical and clinical evidence overcoming many of the shortfalls of conventional chemotherapeutics. Multiple drugs have been attached to cell-penetrating peptides in preclinical research in an attempt to target tumors. Five classes in particular have shown great promise: conventional chemotherapeutics, pro-apoptotic peptides/proteins, NP formulated peptides, antitumor antibodies and siRNA. Co-administration of a cell-penetrating peptide with a tumor-targeting drug allowed the drug to penetrate into tumor’s extravascular space in a tumor-specific and neuropilin-1-dependent manner [54]. Interestingly enough, peptide coadministration with a wide variety of tumor-targeting drugs, such as small molecule chemotherapeutics, NPs, and monoclonal antibodies, have all had their therapeutic indices increased as a result of this coadministration [54,55,56]. Thus, the increased tissue permeability seen with these tumor-targeting drugs co-administered with cell-penetrating peptides is a viable strategy for overcoming the poor penetration of currently available chemotherapeutics.

Another promising area of development in cell-penetrating peptides is their apparent ability to overcome drug resistance issues previously seen with modern anticancer drugs. iRGD is a cyclic 9 amino acid residue peptide that was identified using in vivo screening of phage display libraries in mice with tumors [54]. This peptide has unique tumor-homing abilities that has led it to be a candidate for antitumor drug delivery [54]. In one study, the peptide iRGD was administered with nab-paclitaxel, an albumin-bound form of paclitaxel, leading to effective treatment of a previously resistant breast cancer xenograft [54]. While an exact mechanism by which CPP co-administration reduces drug resistance is unknown, it is hypothesized that co-administration could result in drugs entering via endocytosis rather than by classical mechanisms through the cell membrane, resulting in a greater amount of cellular uptake of the chemotherapeutic [49,57]. Furthermore, due to the excellent targeting and transduction of drugs attached to CPPs, a lower systemic drug dose can be used so that the body’s natural mechanisms that induce resistance are less of a factor [58]. Modern anticancer chemotherapeutics lack targeting specificity, which often results in side effects such as nausea, insomnia, bone marrow depression, fatigue, weakness, and many other adverse effects [59]. All malignant tumor masses contain certain molecular markers that are not expressed in normal cells or are expressed at significantly lower rates [60]. Receptors such as IL-11Rα, GRP78, EphA5 have been found to be differentially overexpressed in tumor cells and thus make attractive candidates for targeting [61,62]. By coupling an anticancer drug to a targeting ligand for these receptors, the drug can effectively accumulate in the tumor, leading to greater therapeutic effect and fewer side effects [60,62].

Remarkably, recent research suggests that drug delivery with cell-penetrating peptides may also occur via a second general mechanism known as the bystander effect [63]. This pathway allows for co-administration of a drug payload without the drug actually being covalently attached to the cell-penetrating peptide [63]. The pathway involved in this bystander effect is known as the C-end Rule (CendR) pathway, which is an endocytic transport pathway related to but distinct from micropinocytosis [63]. iRGD has been shown to be the peptide that activates this pathway [63]. This peptide binds to a tumor-specific receptor, after which it is proteolytically cleaved and then binds to a second receptor, neuropilin-1, resulting in activation of the CendR pathway [63]. The endocytic vesicles that are formed in the CendR pathway are large and can contain and hence transport a large amount of extracellular fluid [63], including chemotherapeutic drugs present in the interstitium. This phenomenon of the CendR pathway explains why certain peptides can transport drug payloads that are simply co-administered with the peptide and not necessarily covalently attached to the peptide. Many preclinical trials have utilized iRGD and the CendR pathways to deliver anticancer drugs. In one study, a wide variety of anticancer drugs were co-administered with iRGD and all saw an increase in their therapeutic indices [54]. In another study, the anticancer effects of gemcitabine were enhanced in a murine pancreatic cancer model that overexpressed neuropilin-1 when co-administered with Irgd [64]. Similar results have been seen in studies examining gastric cancers and hepatocellular carcinoma [65,66]. These successful preclinical trials that co-administered anticancer drugs with iRGD validate the CendR pathway and provide a basis for future clinical research.

Conventional chemotherapeutics have shown promise when attached to cell-penetrating peptides. Paclitaxel is a widely used chemotherapeutic. Paclitaxel can stop mitosis and cause cell death by binding to microtubules [67]. As mentioned earlier, many conventional chemotherapeutics, such as paclitaxel, have issues such as poor solubility, toxicity and acquired resistance. Conjugating paclitaxel to various cell-penetrating peptides has proven to be advantageous in minimizing these negative aspects and maximizing therapeutic effects [68,69,70,71]. Of particular interest is the recent work in attaching octa-arginine to the C2′ position of paclitaxel via a disulfide linker [70,71]. When attached to octa-arginine, paclitaxel was able to overcome drug resistance, and increase solubility [70,71]. In mice with ovarian cancer the octa-arginine-paclitaxel conjugate had a 4.8-fold higher therapeutic response compared with paclitaxel alone [71]. Another commonly used chemotherapeutic, doxorubicin, has also shown promising results when attached to a CPP. Doxorubicin is a topoisomerase II inhibitor that has been used clinically to treat many types of cancer [72], but faces many of the same issues as paclitaxel. Doxorubicin has an intrinsic P-glycoprotein overexpression, which makes it particularly difficult to obtain effective therapeutic tumoral concentrations and can cause resistance [73,74]. In one study, a Tat–doxorubicin conjugate was designed and administered to resistant KB-V1 tumor cells. The results showed 86% of tumor cell cytotoxicity with the Tat–doxorubicin conjugate versus only 14% with doxorubicin alone [75].

Pro-apoptotic proteins have also shown preclinical promise when attached to CPPs particularly p53 [76,77,78]. 11 poly-arginine peptides (11R) have also been shown to suppress the proliferation of oral cancer [76]. The conjugate 11R–p53 suppressed activity of the p21/WAF promoter thus stopping the proliferation of cancer cells [76]. Further studies have shown that linking the polyarginine-p53 fusion protein to the NH2-terminal of a modified influenza virus subunit was able to inhibit the proliferation of bladder cancer [77,78]. Thus, p53 as a pro-apoptotic protein has shown great promise when attached to CPPs, leading to halting proliferation of many different types of cancer cells.

Monoclonal antibodies have long been used in medicine particularly in immunotherapy. However, one of the main concerns with using monoclonal antibodies in oncology has been the issues with cell penetration due to their large size (150 kDa) [79]. By attaching these antitumor antibodies to CPPs, studies have shown promise in overcoming this cell membrane barrier. The cell-penetrating antibody 3E10 recognizes and physically binds to the N-terminus of RAD51 subsequently sequestering it in the cytoplasm and preventing it from binding to DNA and causing damage leading to cancer [79]. Another cell-penetrating antibody, RT11, has been shown to be internalized and selectively bind to activated GTP-bound form of oncogenic Ras mutants which blocks downstream signaling of these mutants and prevents the proliferation of tumor cells [80]. Other research has shown great promise in designing cell-penetrating antibodies with high cell-specificity and high endosomal escape efficacy such as epCT65 which has great potential for medical applications such as cystolic delivery of drug payloads to tumors [80]. There are currently 12 FDA-approved antibodies used for treating cancer and despite their promise in preclinical research when attached to CPP, there are still issues with them such as solubility and intracellular stability [78].

Antibodies formulated with CPPs into NPs are another promising strategy since they have increased solubility and intracellular stability compared to antibodies alone [79,81]. CapG is part of the actin filament, often overexpressed in breast cancer, and is believed to play a role in tumor cell metastasis [82]. Attaching a nanoparticle that works against CapG to various CPPs has been shown to be an effective strategy in reducing breast cancer metastasis [81,82]. More recent research suggests that just the presence of NPs in addition to a CPP and antitumor drug enhances the effect of the latter [83]. One study showed that the delivery of a pro-apoptotic drug as part of a NP-CPP system increased antitumor activity by a factor of 100–300 [83]. This system has shown incredible promise in treating glioblastoma in preclinical research [83]. Interestingly enough, attaching the CPP iRGD to NPs increased its antimetastatic activity as compared to when it is just used as a soluble peptide [78,84].

Small interfering RNA (siRNA) are non-coding RNAs that stop the expression of certain genes by degrading mRNA created during translation. This ultimately prevents ribosomes from translating the mRNAs into functional proteins [85,86,87,88,89]. SiRNAs have shown great potential at treating cancer, but their adaption in clinical settings has been slow due to a lack of safe and effective vehicle for delivery [85]. SiRNAs are intrinsically susceptible to degradation by enzymes, endosomal entrapment and poor cellular uptake [85,86,87,88,89]. Various types of delivery vehicles have been tested to deliver siRNA such as lipids, cationic polymers, inorganic materials and many others; however, synthetic polypeptides were shown to be the most effective [85]. CPPs have shown promise in preclinical research as safe and effective vehicles for siRNA to target tumors. Many peptides, such as CPP33, gh625, PD-L1, and PEG-SS-PEI, have been used in combination with siRNA to target various forms of cancer to great success in animal models [85,86,87,88,89]. Not only did these peptides coupled to siRNA increase the ability to enter tumor cells, but they helped aid in endosomal escape of the siRNAs once they were internalized into cells [85,86,87,88,89]. CPP33 loaded with siPLK1 (an siRNA to target A549 lung cancer cells) exhibited additional endosomal escape but also prolonged blood circulation, enhanced tumor accumulation and effective suppression of tumor growth [85]. Additionally, coupling siRNA to cell-penetrating peptides reduced the concentration of siRNA required to achieve reduction in tumor size [88]. These CPPs with their siRNA payloads have shown enhanced antitumor effects in multiple types of cancer in mice including lung, breast and liver cancers [85,86,87,88,89]. Thus, conjugating siRNAs to CPPs offers another promising avenue in clinical research for the overall treatment of cancer.

## 4. Clinical Trials Using CPPs as Cancer Therapeutics

The authors currently know of ten clinical trials involving CPPs to treat cancer. This number is expected to increase as additional advancements are made in the arena of targeted cancer therapeutics. Of the ten clinical trials discussed in the table below (Table 1), six have completed at least Phase 1a and are continuing onto Phase 1b and Phase 2; the other four are in the process of recruiting and completing Phase 1. Aileron therapeutics appear to be a leader in innovation with their ALRN-6924 peptide that is being successfully employed in half of the clinical studies discussed below. ALRN-6924 is a CPP that disrupts interaction between p53 tumor suppressor protein and its inhibitors MDMX and MDM2 [90,91,92,93,94]. This peptide has been tested alone for safety and efficacy as well as combined to many anticancer drugs such as cytarabine, paclitaxel and topotecan [90,91,92,93,94]. Phase 1 clinical trials have shown that ALRN-6924 alone and in combination is safe and increases the therapeutic index of the covalently attached drugs [90,91,92,93,94]. ALRN-6924 is currently being employed in two clinical trials that are in Phase 2 and have shown promising results in cancer treatment including pediatric cases [93,94]. Another peptide currently in clinical trial is BT1718, designed to target and inhibit the function of MT1-MMP by recognizing and attaching itself to the MT1-MMP protein [95]. Once it is attached it is internalized into cancer cells [95]. P28 is another CPP being evaluated currently in two cancer clinical trials. It is derived from azurin and targets solid tumors that resist standard methods of treatment [96,97]. Both of these trials have completed Phase 1 and look promising at treating solid tumors resistant to conventional chemotherapeutics. PEP-010 is another peptide about to begin enrollment into a Phase I trial to assess its safety profile [98].

## 5. Summary

Since their identification nearly twenty-five years ago, the number and applications of CPPs, both in the arena of tumor diagnostics and therapeutics, continue to grow at a brisk pace. Combining them as novel vectors for targeted delivery of both established and emerging therapeutics has the potential to reduce drug doses, decrease tumor resistance and reduce off-target adverse effects that so often limit dosage of chemotherapeutics, as well as adversely affect patient quality of life. While the future of CPPs in medicine is promising, there are still many issues and challenges that need to be addressed to make their future in medicine feasible. One such challenge will be endosomal entrapment of the peptides; this could potentially be overcome by conjugating drugs or peptides to a peptide that causes endosomal lysis. Another issue to overcome is the human body’s natural immune response and the body’s generation of antibodies that target antitumor drugs. This could be overcome by delivering immunosuppressants with the peptide–drug combination, or finding ways to locally administer the conjugate. Finding a way to deliver peptides successfully to specific organelles within specific types of cells is another challenge. While CPPs have shown promise at delivering cargo to specific cell types, there needs to be more work on targeting specific organelles. One possible solution to this is to add organelle localization sequences to peptides as well as adding endosome-lytic peptides. Finally, proteases are also a concern as they could break down the peptide. This could be overcome by using the D-enantiomers of the amino acid forming the peptide or protecting the CPPs in liposomal or nanoparticle formulations. All of this goes to show that CPPs have immense potential in both cancer diagnostic and therapeutic applications; however, further research is needed for them to become truly efficacious in the field of oncology.

## Figures and Tables

**Figure 1 pharmaceutics-13-00890-f001:**
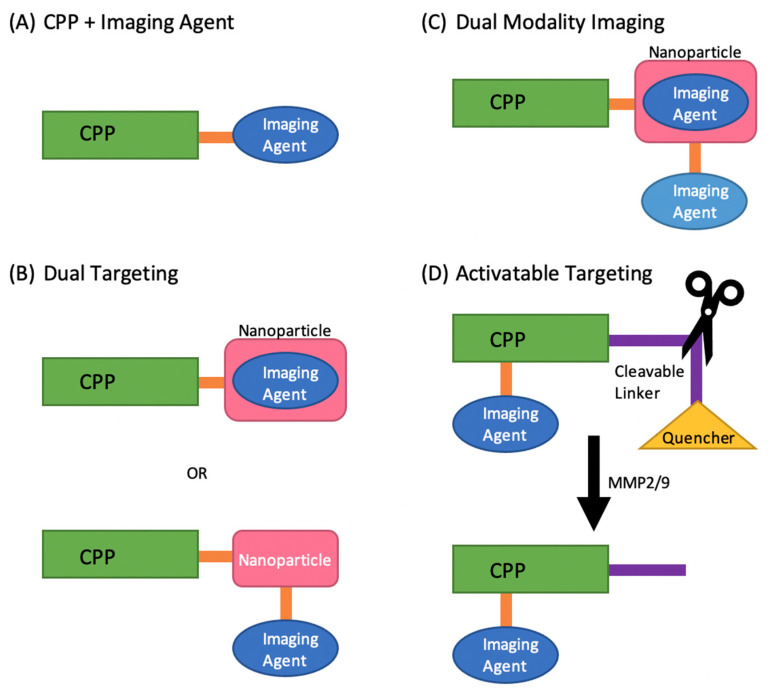
(**A**) CPP + imaging agent directly combines a specific or non-specific CPP with an imaging agent via a linker and/or conjugation. (**B**) Dual targeting combines a CPP with another target-specific component, such as a nanoparticle, via a linker and/or conjugation. An imaging agent can then be linked to the combination or embedded within a nanoparticle. (**C**) Dual-modality imaging combines a CPP with two imaging agents, allowing for different imaging methods. In this case, the imaging agents are attached via a linker and embedded within a nanoparticle. (**D**) Activatable targeting combines CPP and an imaging agent with a MMP2/9-cleavable linker and quencher for the imaging agent. The CPP can only be imaged when MMP2/9 are present to cleave off the quencher.

**Table 1 pharmaceutics-13-00890-t001:** Summary of various clinical trials utilizing CPPs in anticancer therapies.

Sponsor	ClinicalTrials.govIdentifier	Study Stage	CPP Employed	Cancer Targeted	Drug Employed with CPP	Study Size
Aileron Therapeutics [90]	NCT02264613	Phase 1—Completed Phase 2a—Completed	ALRN-6924	Solid tumor, lymphoma, and peripheral T-cell lymphoma	ALRN-6924—alone and in combination withpalbociclib	149
Aileron Therapeutics [91]	NCT02909972	Phase 1—Completed	ALRN-6924	Acute myeloid leukemia, and advanced myelodysplastic syndrome	ALRN-6924—alone and in combination with cytarabine	55
Aileron Therapeutics [92]	NCT03725436	Phase 1	ALRN-6924	Advanced, metastatic or unresectable solid tumors	ALRN-6924—in combination with paclitaxel	45
Aileron Therapeutics [93]	NCT03654716	Phase 1	ALRN-6924	Pediatric leukemia, pediatric brain tumor, pediatric solid tumor, pediatric lymphoma	ALRN-6924—alone or in combination with cytarabine for patients with leukemia	69
Aileron Therapeutics [94]	NCT04022876	Phase 1a—Completed Phase 1b Phase 2	ALRN-6924	Small cell lung cancer	Phase 1b—ALRN-6924 with topotecan Phase 2—topotecan alone and in combination with ALRN-6924	120
Cancer Research UK [95]	NCT03486730	Phase 1 Phase 2	BT1718	Advanced solid tumors, non-small cell lung cancer, non-small cell lung sarcoma, and esophageal cancer	BT1718—alone	130
CDG Therapeutics and Dr. Tapas K. Das Gupta [96]	NCT00914914	Phase 1—Completed	P28	Refractory solid tumors	P28—alone	15
Pediatric Brain Tumor Consortium/National Cancer Institute (NCI) [97]	NCT01975116	Phase 1—Completed	P28	Recurrent or progressive central nervous system tumors	P28—alone	18
Institut Curie [98]	NCT04733027	Phase 1	PEP-010	Metastatic solid tumor cancer	PEP-010—alone PEP-010—in combination with paclitaxel	56
Amal Therapeutics [99]	NCT04046445	Phase 1a—Completed Phase 1b	ATP128	Stage IV colorectal cancer	ATP128—alone and in combination with BI 754091	32

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
