# Peer review of "Cell-Penetrating Peptides: Applications in Tumor Diagnosis and Therapeutics"

_pharmaceutics, 2021, doi:10.3390/pharmaceutics13060890_

Round 1

Reviewer 1 Report

This manuscript reviews the current literature on the use of cell penetrating peptides (CPPs), in particular tumor homing CPPs, for diagnostic and therapeutic applications. This is an interesting topic, and as the authors mention, there are so many important factors to consider in characterizing the properties of these tools.

In general, the manuscript is well written and organized in a logical way. My biggest comment is on the need for another review on CPPs. The authors themselves admit there are already many thorough review articles on the topic (line 62). Their attempt is to focus this one to tumor homing CPPs, and their use in tumor imaging and tumor-specific therapeutics application. However, many of the existing reviews include that information already.

With that said, I believe this manuscript contains useful information and the decision to publish it in light of the current literature should be left to the editor. I just have the following specifics that need to be addressed before publication:

  1. Line 184: please introduce properly (define) “iRGD peptide” (ie move the line 206 up)
  2. There are quite a few typos that should be addressed. For instance, Line 245: please replace “influence virus” by “influenza virus, Line 259: please replace “anti-bodies” by “antibodies”

Author Response

We thank the Reviewer for his/her insightful comments, and thank them for finding it to be well written review. We agree that there are a lot of reviews on CPPs already (some written by us as well), which is why we have narrowed the focus of this review to simply tumor diagnostics and therapeutics. 

We have moved line 206 up and introduced iRGD peptides before delving into it's applications.

We have rectified the grammatical mistakes.

Reviewer 2 Report

Due to the specific ability of breaching the cell membrane barrier while carrying cargoes larger than themselves into cells in an intact, functional form, cell-penetrating peptides (CPPs) have been widely investigated although no CPPs or CPP/cargo complexes have been approved by the US Food and Drug Administration (FDA) yet. This review summarizes the varying strategies which have been developed to enhance their diagnostic potential via derivatizing CPPs for better targeting by constructing specific cell activated forms. The paper is interesting and well written. Minor revision is suggested before possible publication on Pharmaceutics and some suggestions are as follows.

In the first and second sections the authors summarized the studies on CPPs as “tumor imaging agents” or “vectors for targeted drug delivery to tumors”. So many results were listed one by one without any tables or images. For a more direct and accurate impression, images (e.g., on the design and structure of the conjugated molecules or nanoparticles) and tables (summarizing the molecules and strategies) are suggested.

Also, in the last section, in addition to a brief summary of the paper content, more discussions should be involved, such as the advantages and shortcomings of the strategies, future perspectives, and outlook on the CPPs.

Some minor errors, e.g., “than” was written as “then” on line 11, page 1.

Author Response

We thank the Reviewer for his/her insightful comments and time. In response to the comments, we have added a figure (Figure 1) to the review to lay out the diagnostic strategies in more depth. The therapeutic section has a list of clinical trials listed as a table (Table 1) summarizing their status and CPPs/drugs being tested. We believe the addition of a figure has enhanced the quality of the review and we thank the reviewer for this suggestion.

The grammatical mistakes have been corrected.